# Antioxidant and Anti-Inflammatory Properties of Recombinant *Bifidobacterium bifidum* BGN4 Expressing Antioxidant Enzymes

**DOI:** 10.3390/microorganisms9030595

**Published:** 2021-03-13

**Authors:** Zhaoyan Lin, Seockmo Ku, Taehwan Lim, Sun Young Park, Myeong Soo Park, Geun Eog Ji, Keely O’Brien, Keum Taek Hwang

**Affiliations:** 1Department of Food and Nutrition, and Research Institute of Human Ecology, Seoul National University, Seoul 08826, Korea; zy-1221@snu.ac.kr (Z.L.); imtae86@snu.ac.kr (T.L.); sunyoung.park@snu.ac.kr (S.Y.P.); 2Fermentation Science Program, School of Agriculture, College of Basic and Applied Sciences, Middle Tennessee State University, Murfreesboro, TN 37132, USA; seockmo.ku@mtsu.edu (S.K.); keely.obrien@mtsu.edu (K.O.); 3Research Center, BIFIDO Co., Ltd., Hongcheon 25117, Korea; bifidopark@bifido.com (M.S.P.); geji@snu.ac.kr (G.E.J.); 4BK21 FOUR Education and Research Team for Sustainable Food & Nutrition, Seoul National University, Seoul 08826, Korea

**Keywords:** recombinant bifidobacteria, catalase, superoxide peroxidase

## Abstract

*Bifidobacterium bifidum* BGN4-SK (BGN4-SK), a recombinant strain which was constructed from *B. bifidum* BGN4 (BGN4) to produce superoxide dismutase (SOD) and catalase, was analyzed to determine its antioxidant and anti-inflammatory properties in vitro. Culture conditions were determined to maximize the SOD and catalase activities of BGN4-SK. The viability, intracellular radical oxygen species (ROS) levels, intracellular antioxidant enzyme activities, and pro-inflammatory cytokine levels were determined to evaluate the antioxidant and anti-inflammatory activities of BGN4-SK in human intestinal epithelial cells (HT-29) and murine macrophage cells (RAW 264.7). Antioxidant enzymes (SOD and catalase) were produced at the highest levels when BGN4-SK was cultured for 24 h in a medium containing 500 μM MnSO_4_ and 30 μM hematin, with glucose as the carbon source. The viability and intracellular antioxidant enzyme activities of H_2_O_2_-stimulated HT-29 treated with BGN4-SK were significantly higher (*p* < 0.05) than those of cells treated with BGN4. The intracellular ROS levels of H_2_O_2_-stimulated HT-29 cells treated with BGN4-SK were significantly lower (*p* < 0.05) than those of cells treated with BGN4. BGN4-SK more significantly suppressed the production of interleukin (IL)-6 (*p* < 0.05), tumor necrosis factor-α (*p* < 0.01), and IL-8 (*p* < 0.05) in lipopolysaccharide (LPS)-stimulated HT-29 and LPS-stimulated RAW 264.7 cells compared to BGN4. These results suggest that BGN4-SK may have enhanced antioxidant activities against oxidative stress in H_2_O_2_-stimulated HT-29 cells and enhanced anti-inflammatory activities in LPS-stimulated HT-29 and RAW 264.7 cells.

## 1. Introduction

The bioactivity of probiotics has been improved by gene transfer and recombination processes. Especially, recombinant probiotics capable of producing antioxidant enzymes such as superoxide dismutase (SOD) and catalase have been developed to enhance their performance in various applications [1]. Antioxidant enzymes as vital members of antioxidant defense system are deficient in obligate anaerobic bacteria including bifidobacteria [2]. Recombinant probiotics have shown promising results in pre-clinical studies to treat inflammatory bowel diseases by relieving oxidative stress and protecting against cellular damage [3]. Carmen et al. observed that genetically modified strains of *Streptococcus thermophilus* CRL 807 producing catalase reduced the severity of colitis in a 2,4,6-trinitrobenzene sulfonic acid (TNBS)-induced colitis mouse model [4]. Another experimental study reported that recombinant *Lactobacillus casei* BL23 expressing catalase or SOD reduced TNBS-induced Crohn’s disease in mice [5].

*Bifidobacterium* is regarded as a representative probiotic microorganism due to its beneficial effects such as improvement of physiological functions, amelioration of inflammation, and resistance against pathogenic infections [6,7,8,9]. Since bifidobacteria and lactic acid bacteria (LAB) commonly reside in the large intestine and small intestine, respectively [2], recombinant bifidobacteria with antioxidant properties can effectively reduce oxidative stress in the large intestine and treat ulcerative colitis, an inflammatory disease of colon. A previous study reported that recombinant *Bifidobacterium longum* HB25 capable of producing SOD reduced dextran sulfate sodium-induced ulcerative colitis in mice [10].

However, bifidobacteria have some known limitations such as strict anaerobic metabolism, multilayered complex cell walls, and restriction–modification systems. Because of these limitations, most recombinant probiotics have used LAB as host strains for delivery vectors, while recombinant bifidobacteria have been only developed in recent years [11]. In our previous study [2], we optimized the electroporation conditions for *Bifidobacterium bifidum* BGN4 (BGN4) to improve the electroporation-mediated transformation efficiency and constructed recombinant *B. bifidum* BGN4-SK (BGN4-SK) by introducing the SOD gene (*StSodA*) and the catalase gene (*LpKatL*) into BGN4. To our knowledge, only a few engineered bacteria co-expressing SOD and catalase have been developed so far [2].

In this study, further in vitro experiments with BGN4-SK were conducted to verify the bio-functionalities of the newly transformed bacterial strain. Specifically, the conditions for the maximum expression of antioxidant enzymes in BGN4-SK were determined. The antioxidant activity of BGN4-SK against H_2_O_2_ -stimulated oxidative stress was evaluated through human intestinal epithelial HT-29 cells. The anti-inflammatory activity of BGN4-SK against lipopolysaccharide (LPS)-stimulated inflammation was evaluated using human intestinal epithelial HT-29 and murine macrophage RAW 264.7 cells.

## 2. Materials and Methods

### 2.1. Optimum Conditions for Antioxidant Enzyme Activities

#### 2.1.1. Culture of BGN4 and BGN4-SK

BGN4 and BGN4-SK were cultured in de Man Rogosa Sharpe (MRS) medium (BD Difco, Sparks, MD, USA) containing 0.05% L-cysteine HCl (Sigma Aldrich, St. Louis, MO, USA) at 37 °C for 20 h under anaerobic conditions.

#### 2.1.2. Preparation of Crude Enzyme Extract of BGN4-SK

The suspended bacteria (1 mL) were centrifuged at 16,000× *g* for 5 min at 4 °C, and the pellet was harvested. The pellet was washed twice with 50 mM phosphate buffer (pH 7.0). The washed bacteria were suspended in 1 mL phosphate buffer and disrupted with a sonicator (Q500, K-Corporation, Suwon, Korea) for 10 min with 1.0 s on and off. The disrupted suspension was centrifuged at 16,000× *g* for 10 min at 4 °C, and the supernatant was used as crude enzyme extract to assay SOD and catalase activities.

#### 2.1.3. Measurement of SOD and Catalase Activities

SOD activity was measured using an SOD activity assay kit (Dojindo, Kumamoto, Japan). 

Catalase activity was evaluated based on the method described by Park [2]. The dichromate/acetic acid reagent was prepared by mixing 5% K_2_Cr_2_O_7_ with glacial acetic acid (1:3, *v/v*). The crude enzyme extract (1 mL) was mixed with a solution containing 4 mL of 0.2 mM H_2_O_2_ and 5 mL of 10 mM phosphate buffer (pH 7.0), followed by incubation at 40 °C for 5 min. Then, 2 mL of dichromate/acetic acid reagent was added to 1 mL of the reaction mixture. The mixture was heated at 100 °C for 10 min to stop the reaction, and optical density (OD) was measured at 570 nm using a microplate reader (Model 680, Bio-Rad, Hercules, CA, USA).

#### 2.1.4. Optimization of Medium for SOD and Catalase Production by BGN4-SK

SOD and catalase of BGN4-SK are Mn-dependent and heme-dependent enzymes, respectively [2]. Therefore, cofactors (hematin (Sigma Aldrich) and MnSO_4_ (Samchun Chemical, Pyeongtaek, Korea)) should be added to the medium in which the recombinant bacteria are cultured. In our previous study, Park found that BGN4-SK presented the highest catalase activity in MRS medium supplemented with 30 μM hematin [2]; in this study, the optimal concentration of MnSO_4_ was determined. To evaluate the effects of different carbon sources on the enzyme activities, glucose in the standard medium was substituted with lactose or fructose at 2% (*w/v*). After that, the optimal incubation time in different carbon sources was also investigated.

### 2.2. Preparation of Bacteria Culture and Cell Culture and Cell Viability Assay

#### 2.2.1. Culture of BGN4 and BGN4-SK

BGN4 and BGN4-SK were cultured in MRS medium containing 0.05% L-cysteine HCl, 30 μM hematin, and 500 μM MnSO_4_ at 37 °C for 24 h under anaerobic condition.

#### 2.2.2. Cell Lines and Culture Conditions

RAW 264.7 (KCLB 40071) and HT-29 (KCLB 30038) cell lines were purchased from the Korea Cell Line Bank (Seoul, Korea). The cells were cultured in Dulbecco’s modified Eagle’s medium (DMEM, Gibco, Grand Island, NY, USA) supplemented with 10% (*v/v*) heat-inactivated fetal bovine serum (Gibco) and 1% penicillin/streptomycin (Sigma Aldrich) at 37 °C in an atmosphere of 5% CO_2_.

#### 2.2.3. Cell Viability Assay

3-(4,5-Dimethylthiazol-2-yl)-2,5 diphenyl tetrazolium bromide (MTT, Sigma Aldrich) assay was performed to evaluate the effects of wild-type or recombinant bacteria (5 × 10^8^ CFU/mL) on the viability of RAW 264.7 and HT-29 cells. RAW 264.7 and HT-29 cells were seeded in a 96-well plate at 1 × 10^5^ cells per well and incubated for 24 h with 5% CO_2_ at 37 °C. BGN4 and BGN4-SK were adjusted to 5 × 10^8^ colony-forming units (CFU)/mL and suspended in 100 μL of DMEM without antibiotics. Subsequently, the cells were treated with the bacterial suspension with or without 100 μM (final concentration) H_2_O_2_ aqueous solution or 100 ng/mL (final concentration) LPS (Sigma Aldrich) and incubated for 24 h at 37 °C. After removing the medium, 100 μL of MTT solution (5 mg/mL) was added to the cells, followed by incubation for 4 h. Then, 100 μL of dimethyl sulfoxide (Sigma Aldrich) was added to each well. After 20 min of incubation, the OD was measured at 540 nm using the microplate reader.

### 2.3. Antioxidant Activities of BGN4-SK in H_2_O_2_-Stimulated HT-29 Cells

#### 2.3.1. Determination of Intracellular Radical Oxygen Species (ROS) Levels 

HT-29 cells were cultured in a 96-well plate (black and clear bottom) at a density of 1 × 10^5^ cells per well for 24 h at 37 °C. The cells were treated with 100 μL of bacterial suspension at 5 × 10^8^ CFU/mL, along with 100 μM (final concentration) of H_2_O_2_ followed by incubation for 24 h at 37 °C. The cells were washed twice with phosphate-buffered saline (PBS). Then 100 μL of 2′,7′-dichlorofluorescein diacetate (DCF-DA, Sigma Aldrich) was added to the cells. After 45 min of incubation in the dark at 37 °C, fluorescence intensities at 485 and 535 nm were measured using a multi-mode microplate reader (SpectraMax iD3, Molecular Devices, San Jose, CA, USA).

#### 2.3.2. Determination of Intracellular Antioxidant Enzyme Activity

HT-29 cells were cultured in a 24-well plate at a density of 5 × 10^5^ cells per well for 24 h. The cultured cells were treated with the bacteria (5 × 10^8^ CFU/mL) for 24 h. The cells were washed twice with PBS and scraped from the plate using 100 µL of a mixture of radioimmunoprecipitation assay buffer (Biosesang, Seongnam, Korea) and a protease inhibitor cocktail (Biosesang) (1:100, *v/v*). The cell pellets were kept on ice for 30 min, followed by centrifugation at 18,000× *g* and 4 °C for 15 min. The supernatant was then collected, and the activities of antioxidant enzymes (SOD, catalase, and glutathione peroxidase (GPx)) were determined using an SOD assay kit, a catalase assay kit (Biomax, Seoul, Korea), and a GPx assay kit (Cayman Chemical, Ann Arbor, MI, USA), respectively. Protein concentrations were determined using a modified Lowry protein assay kit (Thermo Fisher Scientific, Waltham, MA, USA).

### 2.4. Determination of Cytokine Levels in LPS-Stimulated HT-29 Cells and RAW 264.7 Cells

HT-29 and RAW 264.7 cells were seeded into a 24-well plate at a density of 1 × 10^6^ cells per well and cultured for 24 h at 37 °C. After removing the medium, the cells were treated with 100 μL of bacterial suspension at 5 × 10^8^ CFU/mL along with 100 ng/mL (final concentration) LPS and incubated for 24 h at 37 °C. The culture supernatant was collected to determine interleukin (IL)-6, tumor necrosis factor (TNF)-α, and IL-8 levels using enzyme-linked immunosorbent assay kits (BD Biosciences, San Jose, CA, USA), according to the manufacturer’s instructions.

### 2.5. Statistical Analysis

Results were expressed as means ± standard deviations (SD). As the data were normally distributed, significant difference among the groups was evaluated with one-way analysis of variance (ANOVA) followed by Duncan’s multiple range test (*p* < 0.05). All statistical analyses were performed using IBM SPSS Statistics 26.0 (Chicago, IL, USA).

## 3. Results and Discussion

### 3.1. Optimum Conditions for the Activities of SOD and Catalase

#### 3.1.1. Effect of MnSO_4_ Concentrations on SOD Activity

The SOD activity of BGN4-SK reached its maximum level when 500 µM MnSO_4_ was added to the medium. However, when the concentration of MnSO_4_ in the medium was higher than 500 µM, SOD activity significantly decreased (*p* < 0.05) (Figure 1). Previous studies have reported that Mn^2+^ influences SOD expression in recombinant bacteria acting as a cofactor and changes SOD activity depending on Mn^2+^ concentration [12,13]. A previous study also reported that addition of Mn^2+^ to Mn^2+^-sufficient medium did not induce further synthesis of manganese superoxide dismutase (Mn-SOD) [14]. Depending on the type of microorganism, achieving Mn-SOD maximum activities requires different Mn^2+^ levels in the medium. Li et al. reported that the SOD activity of recombinant *Escherichia coli* was the highest when 5 mM Mn^2+^ was added to the medium [15]. Another previous study showed that Mn-dependent SOD activity of *Lactobacillus sanfranciscensis* CB1 reached its maximum when 225 µM Mn^2+^ was added to the medium [16].

#### 3.1.2. Effects of Various Carbon Sources and Cofactors on the Growth of BGN4-SK

The growth of BGN4-SK was significantly faster in MRS medium containing glucose, lactose, or fructose as a sole carbon source than in medium containing galactose, sucrose, raffinose, maltose, mannose, arabinose, rhamnose, or xylose (*p* < 0.05) (Figure 2). These results are similar to those of a previous study conducted by Albert et al., indicating that the growth of *Bifidobacterium infantis* strains was higher when glucose, lactose, fructose, or raffinose was used as a sole carbon source [17]. The addition of cofactors had little effect on the growth of BGN4-SK (Figure 2), which is consistent with a previous study indicating that the growth of *L. sanfranciscensis* CB1 expressing Mn-dependent SOD was not affected by the addition of Mn^2+^ [16].

#### 3.1.3. Effects of Carbon Sources on SOD and Catalase Activities of BGN4-SK

The SOD activities of BGN4-SK cultured with glucose and fructose were higher (1.47-fold and 1.32-fold, respectively) than that those of bacteria cultured with lactose (*p* < 0.05) (Figure 3a). The catalase activities of BGN4-SK cultured with glucose and fructose were also higher (1.54-fold and 1.55-fold, respectively) than that with lactose (*p* < 0.05) (Figure 3b). This result is similar to previous studies indicating that the SOD activity of recombinant *E. coli* cultured with glycerol or glucose was relatively higher than that of bacteria cultured with other carbon sources and that the β-galactosidase activity of recombinant BGN4 was higher when glucose or lactose was used as a carbon source [15,18].

#### 3.1.4. Effects of Incubation Times and Carbon Sources on SOD and Catalase Activities

The activities of SOD and catalase in BGN4-SK in the presence of glucose were the highest when the bacteria were cultured for 24 h (Figure 4). Hwang found that the growth of BGN4 reached the early stationary phase at that time [19]. This result is consistent with previous studies indicating that the Fe-dependent SOD activity of a recombinant *Pseudomonas putida* strain was high in the log-growth and stationary phases of bacterial cells grown in a medium supplemented with FeCl_3_ [20] and the β-galactosidase activity of *Lactobacillus leichmannii* 313 was the highest in the early stationary phase [21].

### 3.2. Cell Viability

When BGN4 (5 × 10^8^ CFU/mL) and BGN4-SK (5 × 10^8^ CFU/mL) were incubated with HT-29 and RAW 264.7 cells for 24 h at 37 °C, the cell viability was not affected by the bacterial treatment (*p* > 0.05) (Figure 5a,b). The viability of the HT-29 cells treated with H_2_O_2_ was significantly lower than that of the control cells (*p* < 0.05) (Figure 5c). The H_2_O_2_-stimulated cells treated with the bacteria for 24 h showed suppressed cytotoxicity, resulting in the cell viability becoming similar to that of the control. The viability of the cells treated with BGN4-SK was significantly higher than that of the cells treated with BGN4 (*p* < 0.05). The viability of HT-29 cells and RAW 264.7 cells treated with LPS was significantly lower than that of the control cells, while the viability of LPS-stimulated HT-29 and RAW 264.7 cells treated with the bacteria for 24 h was significantly higher than that of LPS-stimulated cells (*p* < 0.05) (Figure 5d,e). These results are consistent with previous studies suggesting that *Lactobacillus plantarum* ZDY2013 and *B. bifidum* WBIN03 can protect H_2_O_2_-stimulated HT-29 cells from oxidative stress and that the viability of LPS-stimulated HT-29 cells pre-treated with *B. longum* NCC2705 and *B. bifidum* S17 was over 95% [22,23].

### 3.3. Antioxidant Activities of BGN4-SK in H_2_O_2_-Stimulated HT-29 Cells

#### 3.3.1. Intracellular ROS Levels

Compared with the H_2_O_2_-stimulated group, intracellular ROS levels were significantly lower in the cells treated with the bacteria (*p* < 0.05) (Figure 6). The intracellular ROS level of the group treated with BGN4-SK was significantly lower than that of the group treated with BGN4, which showed the same ROS level as the control group (*p* < 0.05). These results are similar to those of a previous study indicating that *E. coli* can successfully reduce oxidative stress through the over-expression of Mn-SOD in response to increased concentrations of intracellular ROS [24].

#### 3.3.2. Intracellular Antioxidant Enzyme Activity

Intracellular SOD, catalase, and GPx activities were measured to evaluate the protective effects of the recombinant bacteria against oxidative stress in H_2_O_2_-stimulated HT-29 cells. Intracellular SOD activity increased 3-fold and 2.47-fold in the BGN4-SK treatment group compared with the H_2_O_2_-stimulated and BGN4 treatment groups, respectively (Figure 7a). Intracellular catalase activity was significantly higher (1.23-fold) in the BGN4 treatment group than in cells treated with H_2_O_2_ (*p* < 0.05). Intracellular catalase activity was significantly higher (2.91-fold) in the BGN4-SK treatment group than in the BGN4 treatment group (*p* < 0.05) (Figure 7b). Furthermore, GPx activity increased 1.46-fold in the BGN4-SK treatment group as compared to the control group (Figure 7c). BGN4 (wild-type) could also improve intracellular antioxidative enzymatic activity. The antioxidant function of probiotics has been demonstrated in investigations of Nrf2 activation and antioxidant gene expression in various probiotic studies [1]. Specifically, *L. plantarum* Y44 stimulated the expression of proteins of the Nrf2 antioxidant pathway, improving catalase activity in 2,2′-azobis(2-methylpropionamidine) dihydrochloride-induced Caco-2 cells [25]. Thus, the increase of SOD and catalase activities in H_2_O_2_-stimulated HT-29 cells treated with BGN4 might be related to the activation of Nrf2 signaling pathway. The results suggest that treatment with BGN4 and BGN4-SK may improve enzymes’ antioxidant activity (Figure 7). These results are consistent with those of a previous study suggesting that the levels of antioxidant enzymes increased in H_2_O_2_-stimulated HT-29 cells treated with *L. plantarum* ZDY2013 and *B. bifidum* WBIN03 [24]. The results of this study indicate that the activities of antioxidant enzymes were high in the cells treated with BGN4-SK, which may decrease the levels of intracellular ROS and then increase cell viability of H_2_O_2_-stimulated cells. SOD and catalase produced by BGN4-SK can decrease oxidative stress, which is consistent with a previous study indicating that recombinant *L. casei* BL23 strains producing SOD or catalase reduced the severity of intestinal inflammation caused by ROS [5].

### 3.4. Anti-Inflammatory Activities of BGN4-SK in LPS-Stimulated RAW 264.7 Cells and HT-29 Cells

#### 3.4.1. Inhibitory Effect of BGN4-SK on the Expression of the Pro-Inflammatory Cytokines IL-6 and TNF-α in LPS-Stimulated RAW 264.7 Cells

LPS significantly stimulated the production of the pro-inflammatory cytokines IL-6 (*p* < 0.01) and TNF-α (*p* < 0.05) in RAW 264.7 cells, compared with the control cells (*p* < 0.01) (Figure 8). Compared with the cells treated with LPS only, IL-6 production was significantly lower in the cells treated with BGN4, while TNF-α production was significantly higher in the cells treated with BGN4 (*p* < 0.05) (Figure 8). This result was similar to that of a previous study indicating that heat-killed *Bifidobacterium* strains (*Bifidobacterium breve* ATCC 15700 and *B. bifidum* BF-1) at 10^8^ CFU/mL increased the production of TNF-α in LPS-stimulated RAW 264.7 cells, and the bacterial strains were inhibitory or stimulatory in TNF-α production depending on the strains and bacterial concentrations [26]. In addition, BGN4-SK suppressed IL-6 and TNF-α production in the cells more significantly than BGN4 (*p* < 0.05) (Figure 8). Previous studies demonstrated that inflammation could be associated with ROS generation, leading to oxidative stress [5]. ROS are known to activate protein kinases and phospholipases that can activate the redox-sensitive transcription factor NF-κB [27]. As a critical inflammatory mediator, NF-κB can promote the transcription of a variety of pro-inflammatory cytokines such as IL-6, TNF-α, and IL-8 [27]. BGN4-SK might reduce LPS-stimulated intracellular ROS accumulation, which may be correlated with the suppression of NF-κB, which regulates the expression of various pro-inflammatory genes [27]. This finding is consistent with a previous study reporting that *B. bifidum* WBIN03 with antioxidant properties decreased the production of the pro-inflammatory cytokine IL-6 in H_2_O_2_-stimulated HT-29 cells [24].

#### 3.4.2. Inhibitory Effect of BGN4-SK on the Expression of IL-8 in LPS-Stimulated HT-29 Cells

IL-8 production was significantly higher in HT-29 cells treated with 100 ng/mL LPS compared with the control cells (*p* < 0.01) (Figure 9). IL-8, a pro-inflammatory cytokine mainly present in neutrophils, can cause a variety of pathophysiological tissue damage [23]. LPS-stimulated IL-8 production in the cells was attenuated by BGN4 (*p* < 0.05) and BGN4-SK (*p* < 0.01). This result is consistent with a previous study indicating that several strains of bifidobacteria, especially *B. bifidum* strains, suppressed IL-8 production in LPS-stimulated HT-29 cells in a dose-dependent manner, blocking NF-κB activation [23]. Moreover, BGN4-SK was more effective in suppressing IL-8 production than BGN4 (*p* < 0.05). Presumably, BGN4-SK suppresses IL-8 production by reducing the molecular signals of ROS.

Previous studies have proved certain properties of BGN4 in vitro and in vivo, such as high colon cell adhesive ability, immunomodulatory capacities, and anti-cancer effects [8]. The in vitro results of this study suggest that the recombinant strain BGN4-SK, into which SOD and catalase genes were introduced, may have enhanced antioxidant and anti-inflammatory activities useful to treat inflammatory diseases. However, the safety of BGN4-SK needs to be evaluated before its use in health-related applications.

## Figures and Tables

**Figure 1 microorganisms-09-00595-f001:**
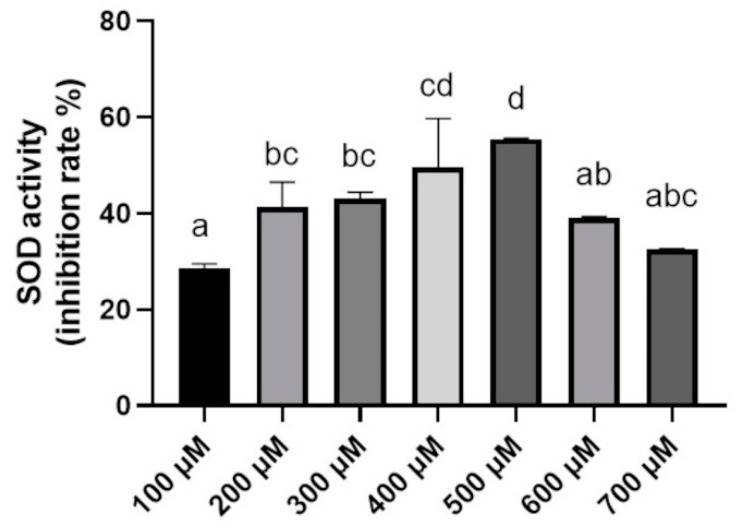
Superoxide dismutase (SOD) activity in *Bifidobacterium bifidum* BGN4-SK cultured in de Man Rogosa Sharpe (MRS) medium supplemented with MnSO_4_ at different concentrations. Data represent means ± standard deviations, *n* = 3. Bars with different small letters indicate significant differences (*p* < 0.05; one-way ANOVA with Duncan’s multiple range test).

**Figure 2 microorganisms-09-00595-f002:**
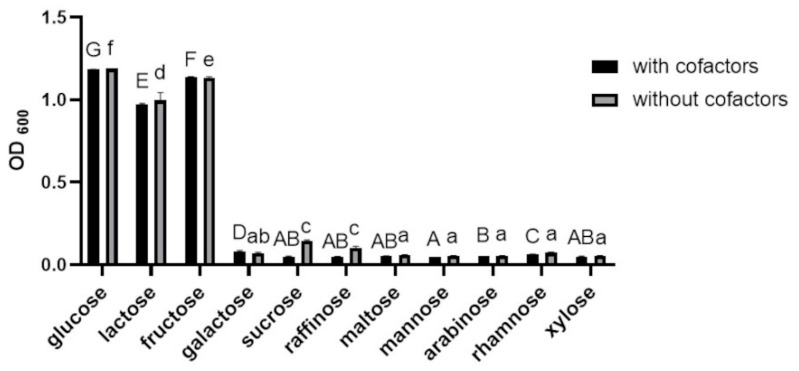
Growth of *B. bifidum* BGN4-SK in MRS medium containing 2% (*w/v*) of different carbon sources with/without cofactors (500 μM MnSO_4_ and 30 μM hematin). Data represent means ± standard deviations, *n* = 3. Bars with different letters indicate significant differences (*p* < 0.05; one-way ANOVA with Duncan’s multiple range test). No significant differences between the samples with and without cofactors (*p* > 0.05; Student *t*-test).

**Figure 3 microorganisms-09-00595-f003:**
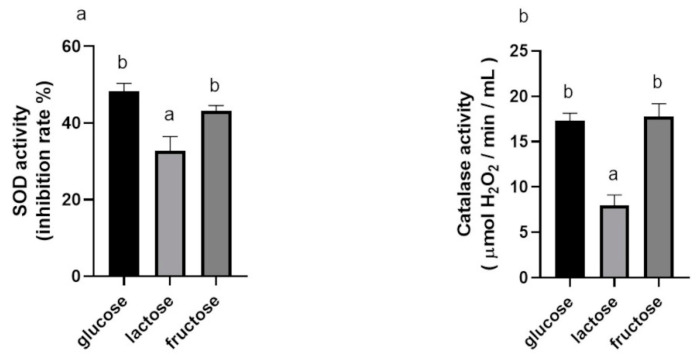
SOD (**a**) and catalase (**b**) activities of *B. bifidum* BGN4-SK in MRS medium containing 2% (*w/v*) of different carbon sources. Data represent the means ± standard deviations, *n* = 3. Bars with different small letters indicate significant differences (*p* < 0.05; one-way ANOVA with Duncan’s multiple range test).

**Figure 4 microorganisms-09-00595-f004:**
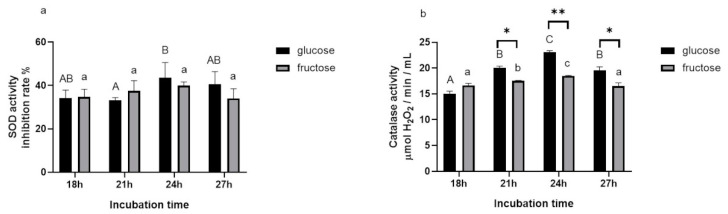
SOD (**a**) and catalase (**b**) activities of *B. bifidum* BGN4-SK at different incubation times and with different carbon sources. Data represent the means ± standard deviations, *n* = 3. Bars with different letters indicate significant differences (*p* < 0.05; one-way ANOVA with Duncan’s multiple range test). Asterisks indicate significant differences in enzyme activity between BGN4-SK in the media containing glucose and fructose (* *p* < 0.05, ** *p* < 0.01; Student’s *t*-test).

**Figure 5 microorganisms-09-00595-f005:**
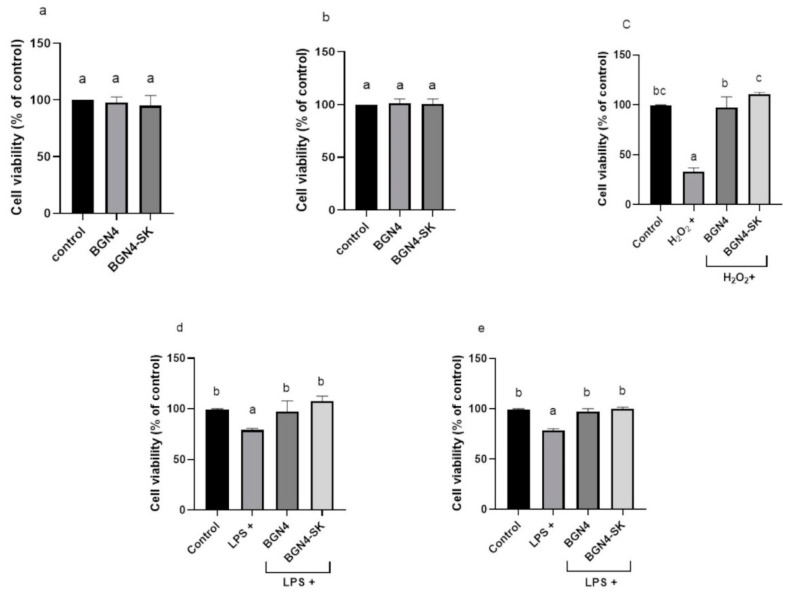
Viability of HT-29 (**a**), RAW 264.7 (**b**), H_2_O_2_-stimulated HT-29 (**c**), lipopolysaccharides (LPS)-stimulated HT-29 (**d**), and LPS-stimulated RAW 264.7 (**e**) cells treated with *B. bifidum* BGN4 and *B. bifidum* BGN4-SK (BGN4 and BGN4-SK, respectively) at 5 × 10^8^ colony-forming units (CFU)/mL. Data represent the means ± standard deviations, *n* = 3. Bars with different small letters indicate significant differences (*p* < 0.05; one-way ANOVA with Duncan’s multiple range test).

**Figure 6 microorganisms-09-00595-f006:**
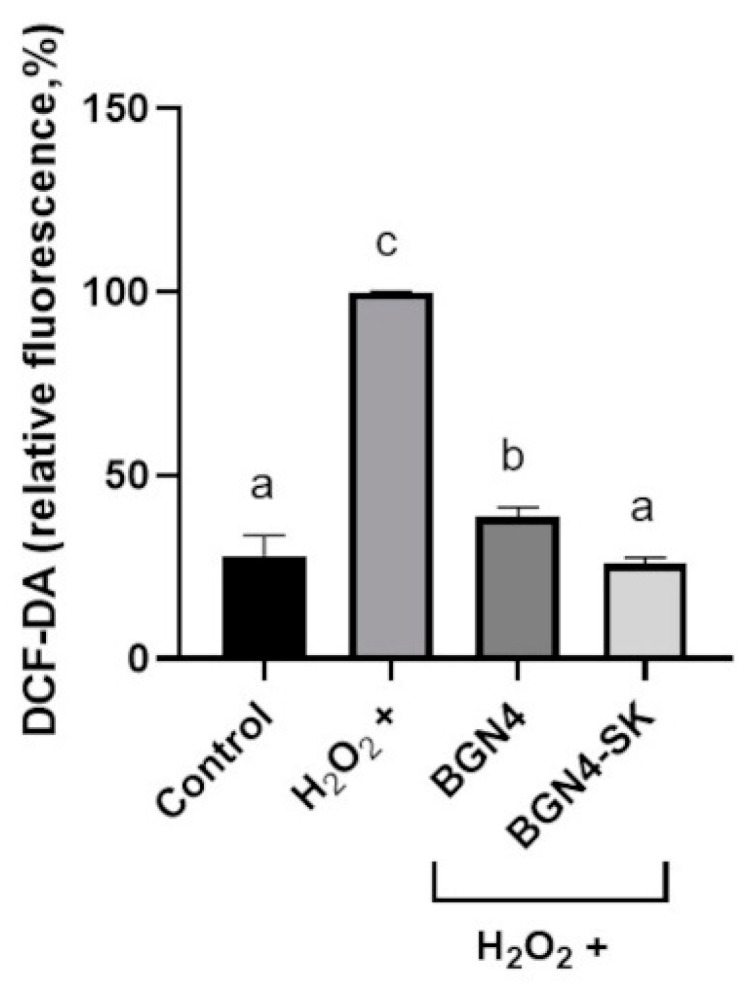
Intracellular reactive oxygen species (ROS) generation in H_2_O_2_-stimulated HT-29 cells treated with BGN4 and BGN4-SK at 5 × 10^8^ CFU/mL. Data represent the means ± standard deviations, *n* = 3. Bars with different small letters indicate significant differences (*p* < 0.05; one-way ANOVA with Duncan’s multiple range test). DCF-DA, 2′,7′-dichlorofluorescein diacetate.

**Figure 7 microorganisms-09-00595-f007:**
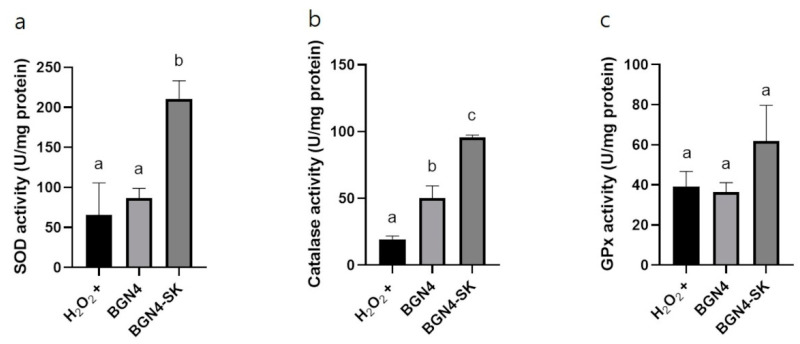
SOD (**a**), catalase (**b**), and glutathione peroxidase (GPx) (**c**) activities in H_2_O_2_-stimulated HT-29 cells treated with BGN4 and BGN4-SK at 5 × 10^8^ CFU/mL. Data represent the means ± standard deviations, *n* = 3. Bars with different small letters indicate significant differences (*p* < 0.05; one-way ANOVA with Duncan’s multiple range test).

**Figure 8 microorganisms-09-00595-f008:**
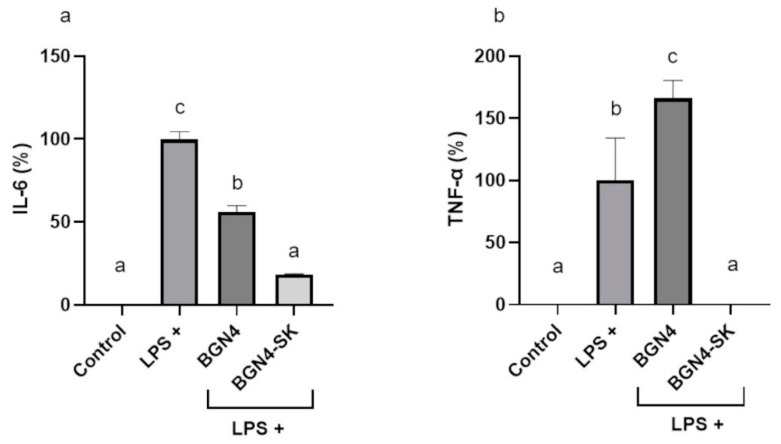
Production of interleukin (IL)-6 (**a**) and tumor necrosis factor (TNF)-α (**b**) in LPS-stimulated RAW 264.7 cells treated with BGN4 and BGN4-SK at 5 × 10^8^ CFU/mL. Data represent the means ± standard deviations, *n* = 3. Bars with different small letters indicate significant differences (*p* < 0.05; one-way ANOVA with Duncan’s multiple range test).

**Figure 9 microorganisms-09-00595-f009:**
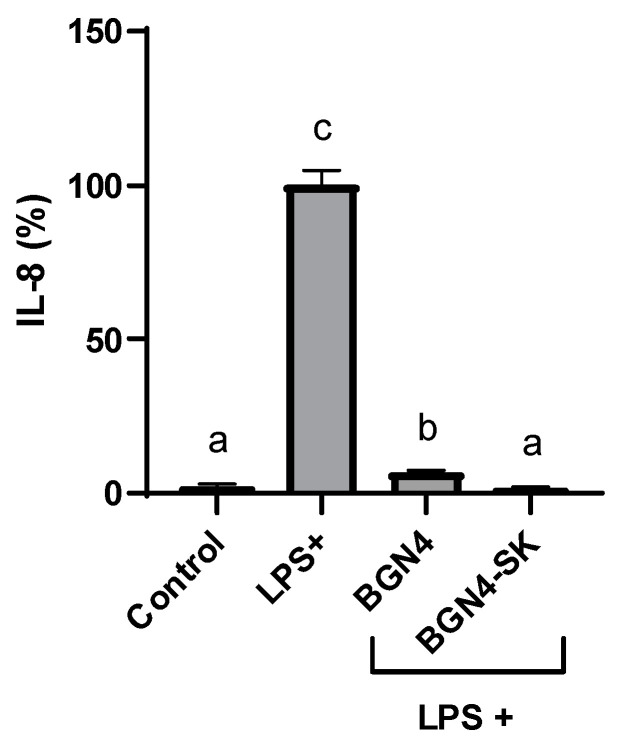
Production of IL-8 in LPS-stimulated HT-29 cells treated with BGN4 and BGN4-SK at 5 × 10^8^ CFU/mL. Data represent the means ± standard deviations, *n* = 3. Bars with different small letters indicate significant differences (*p* < 0.05; one-way ANOVA with Duncan’s multiple range test).

## Data Availability

The data presented in this study are available on reasonable request and for non-commercial purposes.

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
