# Peer review of "Antioxidant and Anti-Inflammatory Properties of Recombinant Bifidobacterium bifidum BGN4 Expressing Antioxidant Enzymes"

_microorganisms, 2021, doi:10.3390/microorganisms9030595_

Round 1

Reviewer 1 Report

Bifidobacterium bacteria are among the dominant components of a healthy bacterial flora. Bifidobacterium has significant probiotic properties. The lactic and acetic acids produced by them lower the pH in the intestinal lumen, thanks to which the growth of pathogenic bacteria is inhibited. In addition, the presence of these bacteria improves the digestibility of certain elements (including iron, zinc, calcium, magnesium). They are used, inter alia, in the prevention of post-antibiotic diarrhea, during H. pylori eradication, as well as in the treatment of irritable bowel syndrome, infectious diarrhea or functional constipation. The Bifidobacterium breve, Bifidobacterium infantis and Bifidobacterium longum bacteria are included in many probiotic supplements. It is an important topic from a global perspective.

Line 16-25: Abstract. Should be improved. As it stands, it is a summary of the authors' previous research. The essence is research and a summary of the research currently carried out. The future reader should be provided factual information related to this manuscript. Therefore, it is unacceptable in its current form and needs to be corrected.

Line 35-72; section: Introduction chapter is briefly described and contains only basic information. In addition, it is largely based on self-information that describes the authors' previous research. More information should be shown in this manuscript in terms of recent research.

Line 86-105; subsection: 2.1.3. Measurement of SOD and catalase activities. Needs some clarification. There are no literature references. The authors are asked to supplement this chapter with literature data. In this case, the manufacturer's instruction number must be stated. If this is a standard recommendation, then this entire section can be deleted.

Line 176-181; subsection: 3.1.1. Effect of MnSO4 concentrations on SOD activity. The authors are asked to describe the obtained results and to interpret in detail in relation to the data presented graphically in Fig. 1.

Line 187-194; subsection: 3.1.2. Effects of various carbon sources and cofactors on the growth of BGN4-SK.
The wording is unclear .. "In the present study, BGN4-SK grew more significantly on glucose, lactose and fructose than on other carbon sources (Figure 2). It was similar to a previous study conducted by Albert et al. (2020), ... "I am asking for a detailed description of the results, obtained results and then refer your results to the existing knowledge and results of other authors. In addition, the statistical analysis of the results should be taken into account in the description. The authors throughout the manuscript do not include the results of the statistical analysis.

Chapter: References. It should be noted that 11 out of 30 cited references are data from 10 years ago. Considering the subject of the article on the basis of preliminary information, it should be considered necessary to refresh the literature so that the experiment can be compared with the latest research.

Author Response

Responses to the Reviewers’ Comments

Reviewer #1

Comment 1: Abstract. Should be improved. As it stands, it is a summary of the authors' previous research. The essence is research and a summary of the research currently carried out. The future reader should be provided factual information related to this manuscript. Therefore, it is unacceptable in its current form and needs to be corrected.

Response 1: We have improved the Abstract of the manuscript as the Reviewer #1 suggested (line 14-30).

Comment 2: section: Introduction chapter is briefly described and contains only basic information. In addition, it is largely based on self-information that describes the authors' previous research. More information should be shown in this manuscript in terms of recent research.

Response 2: We have revised the manuscript more carefully, especially deleting (line 35, 55, 57) and adding (line 62-68) some parts per the Reviewer’s comments.

Comment 3: 2.1.3. Measurement of SOD and catalase activities. Needs some clarification. There are no literature references. The authors are asked to supplement this chapter with literature data. In this case, the manufacturer's instruction number must be stated. If this is a standard recommendation, then this entire section can be deleted.

,

Response 3: We agree with the Reviewer’s comment. Thus, we have deleted the part of SOD measurement since it is a standard method (line 82). We have added literature reference in the part of catalase measurement (line 84).

Comment 4: 3.1.1. Effect of MnSO4 concentrations on SOD activity. The authors are asked to describe the obtained results and to interpret in detail in relation to the data presented graphically in Fig. 1.

Response 4: We have revised the manuscript as suggested (line 152-165).

Comment 5: 3.1.2. Effects of various carbon sources and cofactors on the growth of BGN4-SK.
The wording is unclear .. "In the present study, BGN4-SK grew more significantly on glucose, lactose and fructose than on other carbon sources (Figure 2). It was similar to a previous study conducted by Albert et al. (2020), ... "I am asking for a detailed description of the results, obtained results and then refer your results to the existing knowledge and results of other authors. In addition, the statistical analysis of the results should be taken into account in the description. The authors throughout the manuscript do not include the results of the statistical analysis.

Response 5: We have revised the description of the results in more detail (line 171-179). We have added the statistical analysis of the results throughout the text. We have also revised other parts (line 186-187, 199-200, 282-283) besides the Reviewer’s suggestion in the results.

Comment 6: References. It should be noted that 11 out of 30 cited references are data from 10 years ago. Considering the subject of the article on the basis of preliminary information, it should be considered necessary to refresh the literature so that the experiment can be compared with the latest research.

Response 6: We agree with the Reviewer’s comment that our results should be compared with the recently reported works. We have tried to add more recent references and to delete older ones. However, Bifidobacterium recombinants have been studied even less than E. coli and Lactobacillus recombinants, which have been studied earlier. Therefore, we need to cite some older references (5 out of 27).

Suggestion on English: I don't feel qualified to judge about the English language and style.

Response: One of the authors (Keely O’Brien) is a native English speaker, who has polished English.

Thank the Reviewer #1 for the valuable comments.

Reviewer 2 Report

Very interesting paper but I have a few suggestions: (1) increase the size of Figures 5 and 6 as they are very hard to read and (2) add a Discussion section to expand on the research strengths and limitations and discuss other similar studies in more detail. 

Author Response

Reviewer #2

Comment 1: Increase the size of Figures 5 and 6 as they are very hard to read

Response 1: We have enlarged the size of Figure 5 and 6 as the Reviewer suggested.

Comment 2: Add a discussion section to expand on the research strengths and limitations and discuss other similar studies in more detail.

Response 2: We have expanded the discussion as suggested (line 320-326).

Suggestion on English: Moderate English changes required.

Response: One of the authors (Keely O’Brien) is a native English speaker, who has polished English.

Thank the Reviewer #2 for the valuable comments.
